# Experimental and Numerical Analysis of the Damage Mechanism of an Aramid Fabric Panel Engaged in a Medium-Velocity Impact

**DOI:** 10.3390/polym16131920

**Published:** 2024-07-05

**Authors:** Larisa Chiper Titire, Cristian Muntenita

**Affiliations:** Faculty of Engineering, “Dunărea de Jos” University, 800008 Galati, Romania; larisa.chiper@ugal.ro

**Keywords:** ballistic impact, impact resistance, aramid fabric, body protection

## Abstract

The aim of this study is to analyze the ballistic impact behavior of a panel made of Twaron CT736 fabric with a 9 mm Full Metal Jacket (FMJ) projectile. Three shots are fired at different velocities at this panel. The ballistic impact test procedure was carried out in accordance with NIJ 010106. The NIJ-010106 standard is a document that specifies the minimum performance requirements that protection systems must meet to ensure performance. The 9 mm FMJ projectile is, according to NIJ 010106, in threat level II, but the impact velocity is in threat level IIIA. Analysis of macro-photographs of the impact of the Twaron CT736 laminated fabric panel with a 9 mm FMJ projectile involves a detailed examination of the images to gather information about the material performance and failure mechanisms at the macro- or even meso-level (fabric/layer, thread). In this paper, we analyze numerically and experimentally a panel consisting of 32 layers, made of a single material, on impact with a 9 mm FMJ projectile. The experimental results show that following impact of the panel with three projectiles, with velocities between 414 m/s and 428 m/s, partial penetration occurs, with a different number of layers destroyed, i.e., 15 layers in the case of the projectile velocity of 414 m/s, 20 layers of material in the case of the panel velocity of 422 m/s and 22 layers destroyed in the case of the projectile velocity of 428 m/s. Validation of the simulated model is achieved by two important criteria: the number of broken layers and the qualitative appearance. Four numerical models were simulated, of which three models validated the impact results of the three projectiles that impacted the panel. Partial penetration occurs in all four models, breaking the panel in the impact area, with only one exception, i.e., the number of layers destroyed, in which case the simulation did not validate the validation criterion. The performance of Twaron CT736 fabric is also given by the indentation depth values by two methods: according to NIJ 0101.06 and by 3D scanning. The NIJ 010106 standard specifies that a panel provides protection when the indentation depth values are less than 0.44 mm.

## 1. Introduction

The personal protection system is an over-body covering system intended to safeguard the user during combat engagement [1]. Ancient civilizations invented several types of body protection systems, the use of leather vests being the first such one, and further developed them to assist soldiers by using hard protective covering systems made of bronze and iron [1,2]. The challenge was to provide complete body protection and to simultaneously allow flexible and fast movement in battle. Hard protective covering systems could sometimes lead to an army’s ultimate defeat because of a lack of fast movement.

The development of new materials, using increasingly complex technologies, with a view towards improving the performance, comfort, efficiency, durability, and reliability of armor has been and continues to be a major research aim when it comes to personal safety [3]. In the 1960s, high-strength fibers with high modulus of elasticity were developed, leading to a new era of body armor that provides protection against small arms projectiles to reduce what would otherwise be a fatal wound or a bruise. This development is the result of decades of research and development [4,5,6,7].

The improvement in protection systems has moved from traditional methods to computer-assisted engineering analysis due to the economic and time-saving advantages [8,9,10]. However, challenges still rise when accurately modeling the behavior of materials under extreme conditions, and one of the key issues is the lack of comprehensive material models that consider multi-scale processing in materials, particularly in the case of higher-quality composite materials. Addressing these challenges is important so as to improve the performance of individual protection systems. Numerical simulations of ballistic impact should always be validated by experimental tests [11,12,13,14].

Aramid fiber has exceptional characteristics [15,16,17,18,19,20,21,22,23,24]:It is 43% lighter than glass fiber (density is 1.44 g/cc compared to 2.55 g/cc for glass fiber);It is twice as strong as E-Glass and ten times stronger than aluminum;It offers the same specific tensile strength as high-strength carbon;It has an excellent dimensional stability with a slightly negative coefficient of thermal expansion (−2.4 × 10^−6^/°C);It is resistant to most chemicals except a few strong acids and bases;It can withstand extremely low temperatures down to −320 °F (−196 °C) without loss of strength;It does not melt but begins to carbonize at about 800 °F (427 °C).

Twaron^®^ (Teijin) is a para-aramid material commonly used in protective systems, being five times stronger than steel but flexible, resistant to heat, cutting, and chemicals, and capable of withstanding high ballistic impacts [7,25].

Their unique properties set high-performance synthetic fibers apart from other fibers [26,27]. Ballistic fibers have significantly higher tensile strength and modulus of elasticity, as well as a lower specific fiber strain than any other such material. Ballistic fibers made of polymer are easier to weave than brittle fibers like glass and carbon. Ballistic fibers have proven tolerant to a range of industrial chemicals and solvents [28].

Understanding ballistic hazards is crucial in the design of tough and lightweight body armor. The materials used to create protective systems must meet certain criteria such as the following [29,30,31,32,33,34,35,36,37,38]:Providing sufficient endurance to guarantee wearers adequate protection against ballistic threats;Being light and compact in nature to avoid hindering the wearer’s body movements and effectiveness;Displaying durability, ensuring that its effectiveness does not deteriorate in adverse climatic situations, such as exposure to moisture or UV radiation;Possessing flexibility to provide sufficient comfort;Featuring excellent kinetic energy of absorption and dissipation.

During projectile impact on protective system materials, the following objectives are followed: slowing down the projectile, deforming the projectile as it passes through the panel or layers, stopping the projectile, and reducing the deformation of the back face to reduce trauma to the armor wearer [39,40,41].

NIJ Standard 0101.06 [42] is intended to provide minimum performance requirements and test methods for personal protective equipment used to safeguard from gunfire. Depending on the ballistic protection provided (projectile and initial impact velocity of the projectile), such personal protective systems are classified into five levels (IIA, II, IIIA, III, IV). In addition, there is a special test class, which enables the validation of armor against threats not included in the five standard types. The classification of a protection system providing different levels of protection in different areas of the panel will be determined by the lowest level of ballistic protection provided in any area of the panel.

Denoting the depth of deformation or indentation in the support material (ballistic plasticine) when a projectile hits the protection system, the back face signature in the support material (BFS) is a measure used in ballistic protection system testing to evaluate the performance of protection materials when subjected to ballistic impact [43,44,45,46,47,48,49,50]. Thus, the back face signature is an indicator of the amount of energy transferred to the protection system and it can help determine the ability of the individual protection system to stop or mitigate the effects of a projectile, while minimizing trauma to the human body. In a typical ballistic testing situation, each projectile is subjected to a single back face signature measurement and recording. In the next stage, the recorded BFS values are compared with the limit values set by the standards in [42], in order to evaluate the ability of the protection systems to stop the projectile.

When the back face signature values fall within an acceptable range, the protection system is considered to have passed the test and to have become capable of providing the desired level of protection against the projectile under test. If the back face signature values exceed the specified ranges, the protection system is considered to have failed and does not satisfy the required standards for resistance to a specific projectile.

A 32-ply Twaron CT736 fabric panel was selected for impact testing, a material used because of its superior properties in impact and stab testing, respectively. Based on these results, we chose to test a 32-layer Twaron CT736 panel to evaluate its performance during impact as well. This two-pronged approach is essential for developing versatile and effective protective equipment capable of providing safety in various hazardous scenarios. In the present context, having a frame of reference in place that helps us to validate in one form or another the research carried out, it is possible to demonstrate the relevance and reliability of using 32-layer Twaron fabric panels for ballistic and stab protection.

## 2. Materials and Methods

Intended for lightweight reinforcements (Figure 1), Twaron CT736 is the fabric used in this work and Table 1 provides its characteristics. This panel is composed of two panels of 16 ply, each fastened at the corners, and then fastened with textile shells to form the 32-layer panel. The flexible 32-layer fabric panel was positioned on the bearing material using elastic straps, essential to correctly position the flexible panel during ballistic testing. Elastic straps with hook-and-loop fasteners provide a flexible and adjustable means of securing panels. This allows for consistent and repeatable positioning of armor panels relative to the backing material. The panel is 400 mm × 400 mm which means that conditions specified in the NIJ 0101.06 [42] standard for testing the flexible panel are met. In the range configuration for the testing of protective systems, specific environmental test conditions are set to ensure consistency and accuracy. Environmental conditions, including temperature and relative humidity, are recorded before and after each protective panel firing sequence.

Laboratory testing was carried out in compliance with operating protocols and authorized work instructions, and the weapons and ammunition required for testing were stored in specially constructed and controlled warehouses. According to the NIJ 010106 standard, the trace in the bearing material used in ballistic experiments was measured to an accuracy of ±0.1 mm using a depth gauge. In order to prevent any residue of plasticine on the measurement area, the screwdriver was cleaned after each measurement.

The results are analyzed in keeping with SEM images, the images of the fabric layers, the measurement of the back face signature in the support material by two methods (with the subroll and by 3Dscanning), and the visual analysis of each layer.

Table 2 gives the thickness of the tested panel made of two panels of 16 layers and fastened with textile shells.

The 3D scanning recording and measurement of the indentation depth in the backing material (noted as BFS in NIJ Standard-010106, back face signature) was also analyzed by 3D scanners. One of the most important factors in determining ballistic impact resistance on a flexible vest panel is the depth of the indentation in the backing material. If the projectile has not penetrated the flexible panel, an indentation is formed in the backing material, and the depth of the indentation is a parameter for assessing impact protection. According to the American NIJ Standard-0101.06 [42], this indentation is analyzed by its depth. The proposed measurement method makes a 3D analysis of the indentation in the backing material, including determinations of the area of the indentation in the backing material and its volume.

The panel to be examined under the scanning electron microscope was placed on an aluminum support. The panel was coated (metallized) by a vacuum sputtering process, which consists of applying a very thin layer of gold.

The numerical simulations in this paper are aimed at analyzing the influence of the material properties of the projectile and the yarn, as well as the influence of the initial impact velocity of the projectile. The material properties of the yarn are selected from the literature [52,53,54] as well as the projectile [55,56,57,58,59] and modified to analyze their influence during ballistic impact. The impact between the 9 mm FMJ projectile and the panel made of 32 layers of aramid fabric is simulated in Ansys software 2023. In this study, we have analyzed four variants with the same yarn geometry, projectile, and number of layers:Variant 1, initial projectile impact velocity, 414 m/s, yield strength, 3000 MPa;Variant 2, initial projectile impact velocity, 414 m/s, yield strength, 3600 MPa;Variant 3, initial projectile impact velocity, 422 m/s, yield strength, 3000 MPa;Variant 4, initial projectile impact velocity, 428 m/s, yield strength, 3000 MPa.

The model’s geometry encompasses both projectile and fabric dimensions (Figure 2). Given the meso-level nature of the model, Figure 3 provides the yarn geometry, which closely resembles the actual yarn. The yarn thickness is 0.285 mm. The thickness of the fabric layer provided by the manufacturer is 0.62 mm, the actual (measured) thickness of the fabric layer is 0.57 mm. Both warp and weft yarns share the same cross-section. The size of the simulated panel is 48 mm × 48 mm with two planes of symmetry, and the size of the real panel is 400 mm × 400 mm. The material properties of the yarn and projectile are given in Table 3, Table 4 and Table 5. 

In the paper [60], the authors set up the fabric at the meso-level as well, and the warp and weft yarns have the same cross section. The aim of this paper is to quantitatively analyze (quantitative validation) the energy absorbed by the target.

The meso-level modeling is also addressed in the paper [61], the model being simulated with two planes of symmetry, while the warp yarns and weft yarns have the same cross section.

Characterization of the model:Three initial impact velocities are studied (v_0_ = 414 m/s, v_0_ = 422 m/s, v_0_ = 428 m/s);For the initial projectile velocity of 414 m/s, two values for the yield strength of the yarn (yield strength = 3600 MPa, yield strength = 3000 MPa) are analyzed, and for the initial projectile velocities of 422 m/s and 428 m/s, only one value for the yield strength of the yarn (yield strength = 3000 MPa) is analyzed.Boundary conditions: each thread has fixed lateral cross sections.Contact conditions (frictional contact): frictional contact between yarns with a friction coefficient value of 0.31 and frictional contact between projectile and yarns with a friction coefficient value of 0.31. The friction coefficient is assumed to be constant.The material behavior of the yarn is described using a bilinear model, which suggests a material response characterized by two distinct linear regions.The material behavior of the projectile jacket is described using a bilinear model.The behavior of the projectile core material is described using the Johnson–Cook model.The failure criterion for yarn is Equivalent Plastic Strain (EPS).The failure criterion for the projectile jacket is Equivalent Plastic Strain (EPS).The failure criterion for the projectile core is Equivalent Plastic Strain (EPS).Two planes of symmetry.Mesh = 0.8 mm.

## 3. Experimental Results

### 3.1. Analysis of the Damage Mechanism of Ballistic Impact Panel Using Photographs

The analysis of the macro-level photographs of the impact of panels made of Twaron CT736 stratified fabric with a 9 mm FMJ projectile fabric involves a detailed examination of the images to gather information about material performance and failure mechanisms at the macro-level or even meso-level (fabric/ply, thread).

The ballistic impact analysis based on sample images and SEM images can also be found in [62].

The impact of the three 9 mm FMJ projectiles on the panel made of Twaron CT736 fabric is given in Figure 4. The three projectiles on the flexible panel are fired in the following order: fire B, fire A, and fire C. Fire A, with an initial velocity of 428 m/s, generated a partial penetration into the flexible panel. The higher kinetic energy of projectile A resulted in a more extensive deformation of the fabric structure, but the projectile failed to completely penetrate the panel. Fire B, with an initial velocity of 414 m/s, produced a partial penetration in the Twaron CT736 fabric panel. The velocity of the projectile and its associated kinetic energy caused a localized deformation of the fabric, resulting in the destruction of several layers as follows: 22 layers for fire A (v_A_ = 428 m/s), 15 layers for fire B (v_B_ = 414 m/s), and 20 layers for fire C (v_C_ = 422 m/s).

Figure 4c shows the 3D view of the front face of the flexible panel, which shows that the main yarns have an important role in stopping the projectile. Figure 4d provides a 3D view of the panel’s back face, illustrating the fabric’s behavior during the ballistic impact and its role in absorbing the projectile’s kinetic energy.

Projectile kinetic energy absorption entails four main mechanisms [63]: the energy required to stretch the yarns into pyramids or cones, the energy absorbed by shearing the yarns, the energy absorbed by the movement of the material layers when the layers are stretched and compressed to form the pyramid, and the projectile energy converted into heat.

The weft fibers and warp yarns that form the material layer have a significant impact on their ability to absorb the kinetic energy of the projectile [64].

The ballistic impact results in the transfer of energy from the projectile to the target panel called kinetic energy transfer.

Due to the properties of the target and the parameters of the projectile the following results can be obtained [7,63]:The projectile penetrates the fabric at a certain velocity, which results in an initial projectile energy greater than the amount of energy the target can withstand [7,63].The projectile partially penetrates the fabric layers, resulting in an impact energy of the projectile less than the amount of energy the target could take in. Depending on the material of the target, the projectile may stick to the fabric layers or ricochet [7,63].The projectile completely penetrates the fabric layers with zero residual velocity, resulting in the projectile energy being absorbed by the target [7,63].

Analysis of Fire A

Layer 1 (Figure 5a) shows four partially broken yarns, the fibers that have not been broken are pushed to the side. This effect suggests that the projectile did penetrate between the yarns, causing them to partially break and move laterally. The polymer coating separates from both the impact area and its surroundings. The yarns deformation and the detachment of polymer coating also reveal a difference in the deformation of the two materials, i.e., the aramid fibers that comprise the fabric and exhibit a lower deformation at break, and the polymer film, which exhibits a higher deformation. Layer 16 shows one totally broken yarn and three partially broken yarns. The other main yarns are pushed to the side and have broken fibers. The polymer coating is detached from the fabric over a smaller area than in the previous layers. Layer 32 shows the polymer coating cracking due to the compression of the previous layers.

Analysis of fire B

Layer 1 (Figure 5b) has one totally broken yarn and three partially broken yarns. The other main yarns have broken fibers. Around the impact area, the polymer coating detaches and separates from the surrounding fabric. Layer 16 shows eight stretched and compressed yarns due to projectile impact. This phenomenon of stretched and compressed yarns highlights the complex interaction between the projectile and the layered fabric panel. The strong solicitations exerted by the projectile caused both the drawing of the yarns and the change in their shape by compression. Layer 32 behaves similarly to layer 32 from fire A.

Analysis of fire C

Layer 1 (Figure 5c) has the polymer coating peeled off the fabric in and around the impact area. This layer has four partially broken yarns, and the other main yarns are pushed to the side with broken fibers. Layer 16 shows two totally broken yarns and two partially broken yarns. The complete breakage of the yarns indicates a significant stress and pressure that exceeded their strength limit. The polymer coating behaves similarly to that on layer 15, showing signs of cracking and delamination in and around the impact area. The last layers, 31 and 32, show cracking of the polymer coating due to excessive deformation, even though the projectile was stopped on layer 20.

Although all three shots generated partial penetrations in the Twaron CT736 fabric panel, it is important to point out the differences in material behavior as a function of projectile velocity (Figure 6). The projectiles did not get through the fabric.

Fire A, with a higher initial velocity (v_A_ = 428 m/s), caused more severe localized deformations in the impact area. Twaron CT736 yarns reacted by absorbing and dissipating kinetic forces, attempting to maintain the composite structure’s integrity and reduce the panel’s impact.

Fire B, with a lower projectile velocity (v_B_ = 414 m/s), had a weaker effect on the fabric. Several yarns were destroyed. Even though penetration was not complete, this scenario shows how a high velocity can amplify the destructive effects on the panel.

The appearance of the area damaged by fire C, with an intermediate velocity between fires A and B (v_C_ = 422 m/s), was similar. Partial penetration suggests that the Twaron CT736 fabric structure resisted impact.

Figure 7 shows the layers under the stopped projectile. The yarns are not broken, but in the compression zone of the projectile, some (four yarns, two in the direction of the weft and two in the direction of the warp) are photometrically deformed, and the fact that the impression has been preserved suggests that the yarns are deformed in the plastic do-main). The appearance of the fabric around the projectile indentation suggests a slight twisting of the projectile. The images on the back of these layers show a stretching of the yarns pushed sideways, with a few fibers even being broken. This severe stretching of the laterally pushed yarns is caused by the flattening of the projectile when it is forced to strain because it can no longer move forward in the panel.

The results of these tests provide valuable information for the development and improvement of materials used in ballistic protection or other applications requiring medium velocity impact resistance and efficient kinetic energy absorption. The author of paper [65] mentions that at lower impact velocities, the fabric can absorb more energy because the yarns do not break under initial stress, and transverse deformation can propagate, driving more material into tension.

The authors of paper [66] state that Twaron CT747 aramid fabric shows high protection with advantages of low weight and high mobility.

Koslo et al. [65] analyze a Twaron CT736 fabric panel projectile resistance to impact, hot-pressed, corner-stitched to prevent displacement. Results show that the Twaron CT736 fabric exhibits 0.22 mm caliber projectile resistance in the speed range of 700–1000 m/s.

Shih et al. [23] investigate the impact resistance of aramid fabric reinforced with shear-thickening fluids (STF), epoxies or polyurea elastomers by means of ballistics tests. The results of these tests show that the aramid composite structure treated with polyurea elastomers absorbs the highest amount of impact energy per unit density and provides the best impact resistance. In contrast, the stress concentrator during the ballistic impact decreases the resistance to impact of the epoxy-treated aramid structure. Also, impregnation of the aramid fabric with STF improves structural protection but increases the total weight of the composite structure and reduces the specific energy absorption (SEA). The epoxy-treated aramid fabric shows the best impact resistance. The results were evaluated according to NIJ 010106.

### 3.2. Analysis of the Damage Mechanism of Ballistic Impact Panels Using Scanning Electron Microscopy (SEM)

We performed microscopic examinations of the damaged samples using the scanning electron microscope (SEM), FEI Quanta 200, 4 nm resolution, and 106 magnification power. Figure 8 shows the results of the penetration of the front side of the first layer of the 32-layer Twaron CT736 fabric panel. In Figure 8a, letters A and B mark the yarns that have been completely broken; similarly, letters a, b, and c indicate parts/fragments of the PVB (polyvinyl butyral) polymeric coating. Figure 8b provides a detail that indicates the mode of fiber breakage, which is fibrillated and at stretching, rarely at shearing. The red arrow indicates broken fibers.

In order to explain the damage observed under the scanning electron microscope, Figure 9 and Figure 10, respectively, display failure mechanism characteristics of ballistic impact on fabric panels.

Figure 11 displays the back view of the destruction of the first layer of the panel, consisting of 32 layers of Twaron CT736 fabric. In Figure 11a, letters D and B indicate the yarns that have been completely broken; letters A and C mark partially broken yarns; and letters E and F show yarns pushed to the side with the presence of some broken fibers. In Figure 11b a detail is given to see the mode of fiber breakage, which is fibrillated.

Figure 12a presents a fiber from a main yarn (i.e., from a yarn directly broken by the projectile), and the stress caused the fibrils to detach from each other, and then they were successively broken by stretching. Some ends of the fibrils are twisted and display a large local stretch strain. All these indicate a stretch rupture. In the detail from layer 1 of the 32-layer panel (Figure 12b), three fibers are seen to have been failed by different mechanisms: the top yarn is broken by shearing and crushing; the almost vertical right-hand yarn is twisted and stretched, with some fibrils broken; and the bottom fiber is fibrillated in the zone of breakage, and other fibers are broken by stretching. On the left side, beyond the fibrillation, the fiber is locally gargled, also due to stretching stress.

Figure 12c presents broken fibers at the edge of the penetration orifice. Looking at the micro-level, it is difficult to imagine what happens on a larger scale, but studying the failure mechanisms at the fiber level makes it possible to select the appropriate fibers for a given application. A—sheared fiber (it is observed that the fiber diameter is not modified), B–several fibrils in a fiber have been stretched a lot and have broken one by one, C—fiber broken by stretching with a locally thinned end, typical for polymers, and a little further on, a local twist is seen, D—fiber is broken by shearing, and E—two fibers are broken by stretching and fibrillation.

### 3.3. Analysis of the Indentation Depth in the Support Material

During the scan, various reflective markers were placed on the surface of the de-formed support material for easy detection and recording of the deformation of the sup-port material following a ballistic impact (Figure 13a). Such scanning methods not only provide an accurate result and measurement in a short time, but also facilitate the visual comparison of the indentation depth in the support material.

Using Autodesk Inventor and importing the high-precision records, we were able to visualize the volume of the indentation, calculating the maximum depth of the indenta-tion created in the backing material and the volume of the indentation created in the backing material as a result of the ballistic impact.

Figure 13 shows 3D images of the surface of the bearing material after the panel has been impacted. Figure 14 and Figure 15 show 3D views (front view) and a vertical section through the bearing material in a plane containing the greatest depth of the indentation (which is also named BFS, or indentation depth in the backing material), with the surface of the indentation bounded by a plane containing the undeformed surface of the backing material for the panel under test.

Table 6 gives the BFS results determined by the two measurement methods/instruments. The values obtained were analyzed according to the NIJ 0101.06 standard.

The depth of the indentation, or back face signature, is a measure of how much the material of the personal protective system deforms when impacted by a ballistic threat (in this case, a 9 mm FMJ projectile). The depth of the indentation, or back face signature, plays a crucial role in evaluating the security of the protective system, as it establishes the potential for blunt trauma or injury to the wearer, even if the projectile fails to penetrate the protective system. A shallower back face depth indicates a lower risk of injury because it means that less force is transmitted through the armor to the wearer’s body on impact.

## 4. Analysis of Results from Numerical Simulations

Figure 16 shows equivalent stress images of the simulated variants with three different values of the initial projectile velocity and two different values for the yarn yield strength only in the case of the initial projectile velocity of 414 m/s at the last time point of the simulation, t = 1.5 × 10^−4^ s. Analyzing the four simulated cases, we observe that the projectile is deformed and shows damage over a larger area of the layers when the initial projectile velocity is 414 m/s and the yarn yield strength is 3600 MPa. The three cases, with a 3000 MPa yarn yield strength, show similar damage of the material layers and similar projectile damage of the initial projectile velocities of 414 m/s and 422 m/s. The projectile with an initial velocity of 428 m/s shows more pronounced tip damage compared to the other two projectiles. The initial velocity case of 428 m/s records the highest von Mises stress value at the last moment of the simulation.

The simulated case, with an initial projectile velocity of 414 m/s and a yarn yield strength of 3600 MPa, shows the damage of 17 layers (Figure 17). In this layer, only one yarn was observed to be broken by traction, while the other yarns in the impact zone were subjected to compression. In the case with the yarn yield strength of 3000 MPa and the same initial projectile velocity of 414 m/s, 20 broken layers were recorded. This layer also shows single yarn tensile breaking. The case with an initial projectile velocity of 422 m/s and a yarn yield strength of 3000 MPa shows 20 broken layers. In this case too, only one yarn is broken by tensile stress. The case with an initial projectile velocity of 428 m/s has 21 broken layers. In this case two yarns are broken by traction.

Figure 18a shows the equivalent stress graphs for the three variants studied, with the same yarn yield strength for all three impact velocities. The maximum equivalent stress value recorded at the first time of the simulation (t = 7.5 × 10^−6^ s) for the cases with the same yield strength for the cases with an initial velocity of 414 m/s is for the initial projectile velocity of 428 m/s, with a value of 2781.2 MPa. In the simulated cases with an initial projectile velocity of 414 m/s and 422 m/s, the maximum value of the equivalent stress is 2542 MPa and 2648 MPa, respectively.

In Figure 18b, the graph of the projectile velocity during the simulation is shown for each variant studied. The variation in projectile velocity during simulation is similar for each variant analysis. At the moment of impact, the projectile velocity decreases and the values recorded with a minus sign indicate projectile rebound.

The projectile rebound moment for the studied cases with the same yield strength but different velocities is between the last moment with a positive velocity value and the first moment with a negative velocity value.

In Figure 19a, the equivalent stress graph is shown for the case with an initial velocity of 414 m/s and yarn yield strength values of 3000 MPa and 3600 MPa. Analyzing the graph, we observe that at the beginning of the simulation, the maximum values of the equivalent stress are recorded by the case with the higher yield strength. In the last moments of the simulation, the maximum values for the case with the lower yield strength, while the case with the higher yield strength (of 3600 MPa) maintain a constant stress value starting from the time moment 1.05 × 10^−4^ s.

Figure 19b shows the graph for projectile velocity during impact for the two studied cases of the yarn yield limit, the variant with an initial projectile velocity of 414 m/s. We note that the maximum values are recorded in the case with the lower yarn yield strength.

Figure 20a–f show impact images for the variant with an initial projectile velocity of 414 m/s, 3600 MPa yield strength. The first moment of impact records the damage to a single yarn. Areas with stress concentrators indicate that those areas are prone to being damaged, which is what happened at the next time point. Figure 20g shows the top view of layer 1 at the end of the impact from the numerical simulation, Figure 20h shows the front view of layer 1 from the laboratory test, and Figure 20i the back view of layer 1 from the laboratory test. In the numerical simulation, seven yarns are destroyed, and in the laboratory test, four yarns are destroyed, and the other yarns that are pushed to the side show broken fibers. The yarns that are broken in the numerical simulation are pushed to the side and show broken fibers in the laboratory test. Because during the numerical simulation the meso-level is altered, the yarns are completely damaged. Figure 20i shows the area destroyed by the projectile more clearly (yarns showing broken fibers due to perforation of this layer during impact). This layer shows nine yarns with significant destruction, broken yarns, and a significant number of broken fibers. We can observe a compression in the width of several layers due to the penetration of the projectile through this layer.

Figure 21a–f show impact images for the variant with an initial projectile velocity of 422 m/s and a 3000 MPa yield strength. The first moment of impact records the damage in a single yarn. The experimental simulation records six broken yarns. These yarns are called main yarns because they are in direct contact with the projectile. During impact, the projectile deforms, and the contact area with the yarns increases. Figure 21d shows a twist in the width of the yarn due to the contact of the projectile with the edge of the projectile (deformation of the projectile). The experimental simulation records six broken yarns. These yarns are called main yarns because they are in direct contact with the projectile, the projectile deforms during impact, and the contact area with the yarns increases. Figure 21d shows a twist in the width of the yarn due to the contact of the projectile with the edge of the projectile (deformation of the projectile). Figure 21h very clearly shows the yarns that were pushed sideways by the projectile. In the numerical simulation, these yarns that are pushed to the side are destroyed. Figure 21i illustrates the tensile break mode of the yarns.

Figure 22a–f show impact images for the variant with an initial projectile velocity of 422 m/s, 3000 MPa yield strength. In the numerical simulation, layer 1 shows six broken yarns. Figure 22h shows that seven yarns are damaged by the projectile (total breakage or partial breakage). As in the other cases, it can be seen how the yarns have been compressed across the width due to the penetration of the projectile through this layer. In the numerical simulation, layer 1 shows six broken yarns. Figure 22h shows that seven yarns are destroyed by the projectile (total breakage or partial breakage). As in the other cases, it can be seen how the yarns have been compressed across the width due to the penetration of the projectile through this layer.

## 5. Validation of the Numerical Simulations

Validation of the simulated model is an important step in ballistic protection re-search and development to ensure that the simulation provides plausible and realistic results. The validation of the simulated model is achieved through two important criteria:The number of broken layers (this criterion involves analyzing and recording the number of layers that are broken or damaged during the simulation; this is essential for assessing the effectiveness of the projectile, and its ability to penetrate and damage the target or target structure. The number of broken layers can be an indicator of projectile power and penetration, and is compared with experimental results to validate the model).The qualitative aspect (it involves analyzing the simulation results to check that they correspond to the experimental results; this may include assessing how the projectile behaves in terms of deformation, fragmentation, direction of impact, and any other important characteristics; the qualitative aspect is important to ensure that the simulation correctly reproduces the ballistic phenomena and that the results are plausible).Zochowski et al. [13] validated laboratory tests (Twaron T750 fabric under impact conditions) both qualitatively and quantitatively. The numerical simulation was also modeled at meso-level, with two planes of symmetry, the projectile is 9 mm FMJ, and initial projectile velocity of 460 m/s.Giannaros et al. [67] validated quantitative numerical simulation with laboratory tests of aramid fabrics subjected to ballistic impact. The fabric was modeled at the meso-level. The projectile used is a spherical projectile with a diameter of 12.7 mm and a mass of 8.4 g.Clifton et al. [68] validate laboratory tests on the impact influence of polymer composites made of hybrid and non-hybrid fabrics in terms of quality and quantity (impact energy and absorbed energy). Results obtained from numerical simulations and laboratory tests show differences below 14%.

Figure 23a shows the validation of the qualitative aspect of the variant with an initial projectile velocity of 414 m/s and a yield strength of 3600 MPa, and Figure 23b shows the qualitative aspect of the variant with an initial projectile velocity of 414 m/s, and a yield strength of 3000 MPa. The case with 3600 MPa yield strength shows 17 damaged layers, and the case with 3000 MPa yield strength shows 20 damaged layers. Of these two, only one case satisfies the validation criteria for quality appearance and the validation criteria for the number of damaged layers, namely the case with the 3600 MPa yield strength. The case with the 3000 MPa yield strength shows less deformation of the projectile compared to the laboratory test, and the decrease in the yarn yield strength resulted in more layers being damaged. Figure 23c shows the qualitative appearance of the variant with the initial projectile velocity of 422 m/s and the 3000 MPa yarn yield strength compared to the laboratory test. The numerical simulation shows 20 damaged layers, and the laboratory test shows 20 damaged layers. The case I variant meets both validation requirements.

Figure 24, Figure 25 and Figure 26 show images of different layers from numerical simulations (left) and laboratory tests (right). In the numerical simulations, more yarns are destroyed because the yarn is continuously altered, but in reality it has a large number of fibers. In laboratory tests, yarns that are pushed to the side have broken fibers, whereas in numerical simulations yarns are entirely broken.

These ballistics tests demonstrate how sensitive the stratified material response can be to a relatively small variation in projectile impact velocity, a critical variable. The ability of Twaron CT736 fabric not to be completely perforated is essential in the context of ballistic protection applications, where control of fabric deformation as well as fiber alteration and kinetic energy absorption are of maximum importance. Projectile breakage in the 32-layer Twaron CT736 fabric panel on a given layer (Figure 23) depends on the velocity of the projectile in the sense that for the higher velocity, the projectile penetrated more layers (fire A, 22 layers), for the intermediate velocity, v_C_, 19 layers were broken, and for the lowest impact velocity on this panel, v_B_, the number of layers penetrated was only 14.

## 6. Conclusions

Testing the 32-layer Twaron CT736 fabric panel impregnated with PVB (polyvinyl butyral) according to NIJ 0101.06 has shown that the material can effectively withstand the impact of a 9 mm FMJ projectile with an initial velocity between 414 m/s and 428 m/s.

The tested panel shows partial penetration with a different number of broken layers (15 broken layers for the projectile with an initial velocity of 414 m/s, 20 broken layers for the projectile with an initial velocity of 422 m/s and 22 broken layers for the projectile with an initial velocity of 428 m/s). Based on the existing literature [7,25,63] we can conclude that the obtained experimental results are realistic, and the kinetic energy of the projectile is absorbed by the target.

The panel’s ability to withstand impact is confirmed by the signature of the back face (BFS), a crucial indicator of the material’s performance to absorb energy and disperse im-pact. Measurements carried out in keeping with NIJ 010106 and with 3D scanning of the bearing material, indicate that this signature provides concrete proof of the panel’s integrity and effectiveness in impact.

Using two methods to measure the back face signature—NIJ 010106 and 3D scanning—allows a detailed and accurate assessment of the material behavior under impact conditions. This suggests that the panel has undergone a rigorous and multi-dimensional validation process.

The validation of the simulated model was carried out using two key criteria:Number of broken layers: This is a quantitative indicator of the strength of the material and its ability to distribute the forces generated by the impact. A lower number of broken layers indicates better performance.Qualitative aspect: This provides a visual and structural assessment of the panel after impact, contributing to the understanding of the material behavior and its effectiveness under ballistic stress conditions.

From the numerical analysis, we can conclude that if the value of the yarn yield limit parameter rises, the impact strength of the yarns increases, and ultimately the number of layers with broken yarns decreases. At the same time, a small variation in the initial velocity of the projectile does not influence the number of broken layers.

## Figures and Tables

**Figure 1 polymers-16-01920-f001:**
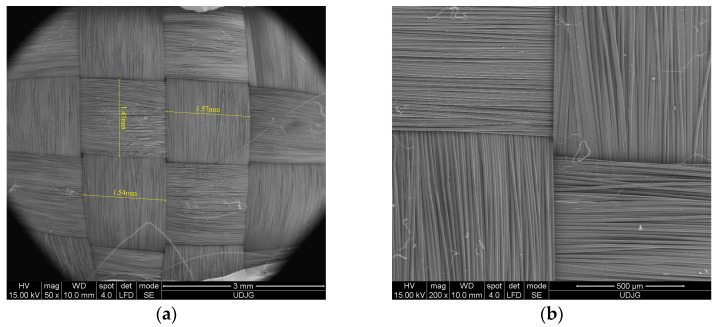
SEM image of the fabric (**a**) yarn dimensions; and (**b**) detail.

**Figure 2 polymers-16-01920-f002:**
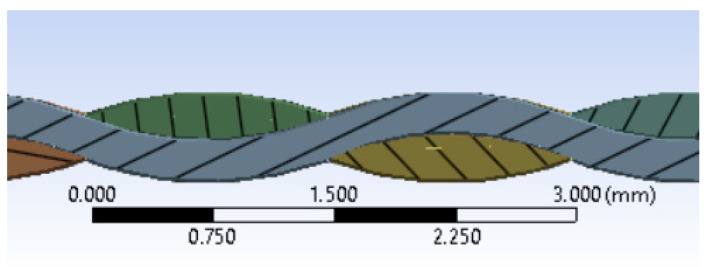
Longitudinal and cross-section of the yarn.

**Figure 3 polymers-16-01920-f003:**
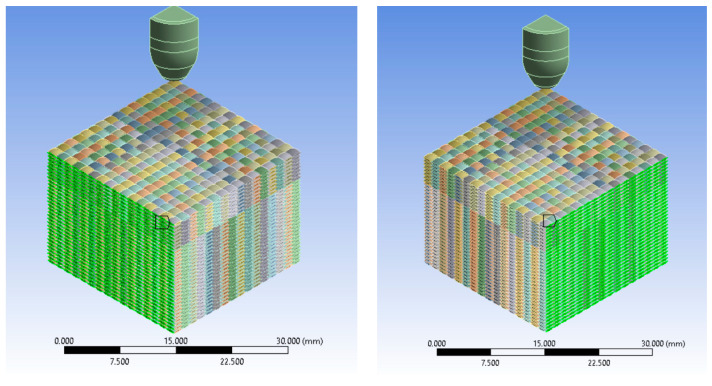
Panel and projectile geometry. Green color indicates the cross-section of the yarn.

**Figure 4 polymers-16-01920-f004:**
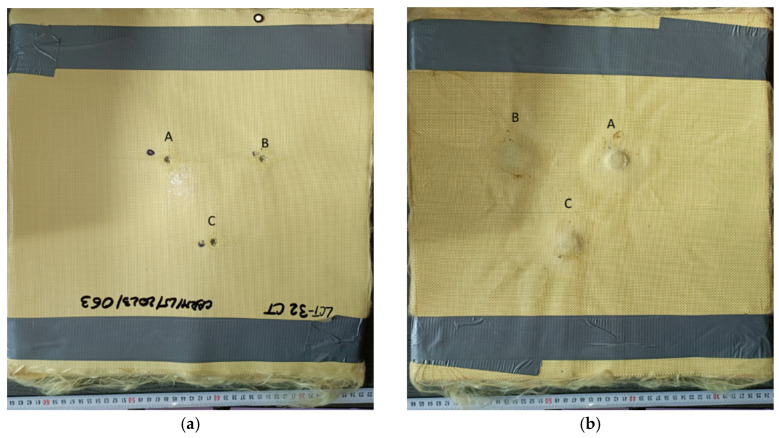
Twaron CT736 32-layer fabric panel after testing: (**a**) front view of the panel; (**b**) back view of the panel; (**c**) front view of the panel by 3D scanning; and (**d**) back view of the panel by 3D scanning.

**Figure 5 polymers-16-01920-f005:**
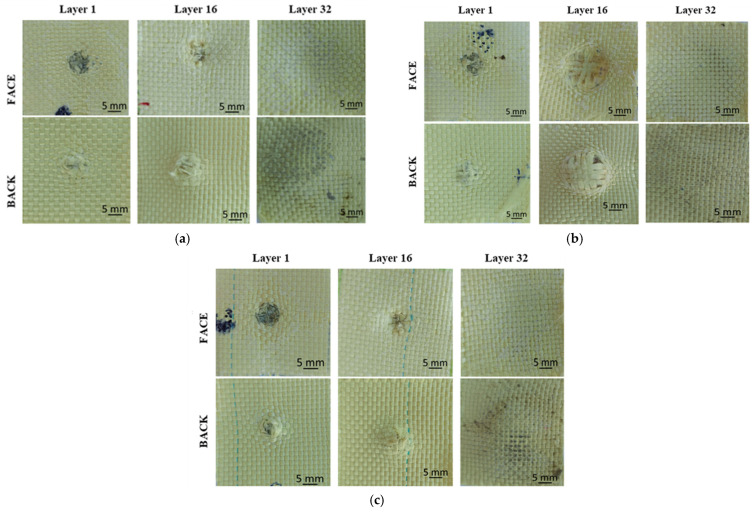
Twaron CT736 32-layer panel after impact: (**a**) fire A; (**b**) fire B; (**c**) fire C.

**Figure 6 polymers-16-01920-f006:**
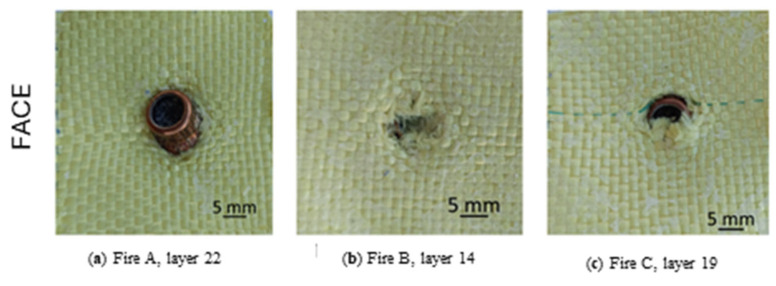
Back view of the layers on which the projectile stopped, for the 32-layer Twaron CT736 panel.

**Figure 7 polymers-16-01920-f007:**
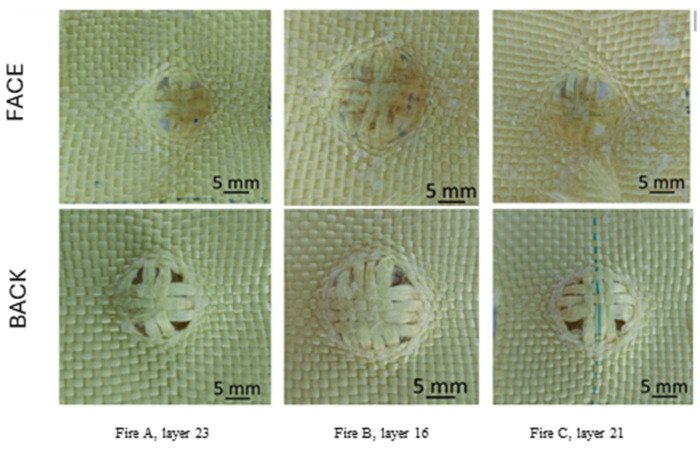
Yarn tensioning on 32-layer Twaron CT736 fabric panel.

**Figure 8 polymers-16-01920-f008:**
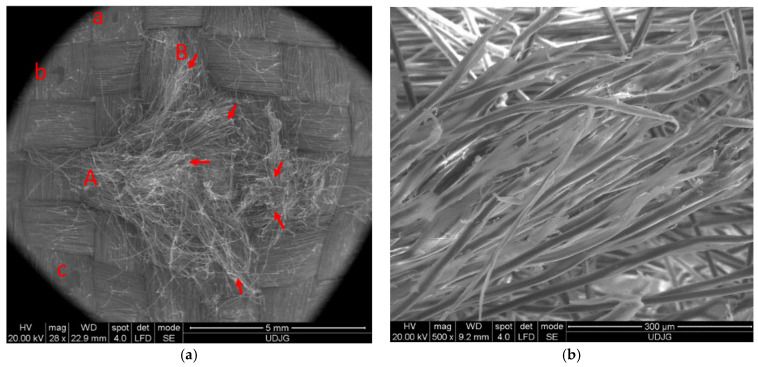
Layer 1 of the panel with 32 layers of Twaron CT736 fabric: (**a**) front view; (**b**) detail.

**Figure 9 polymers-16-01920-f009:**
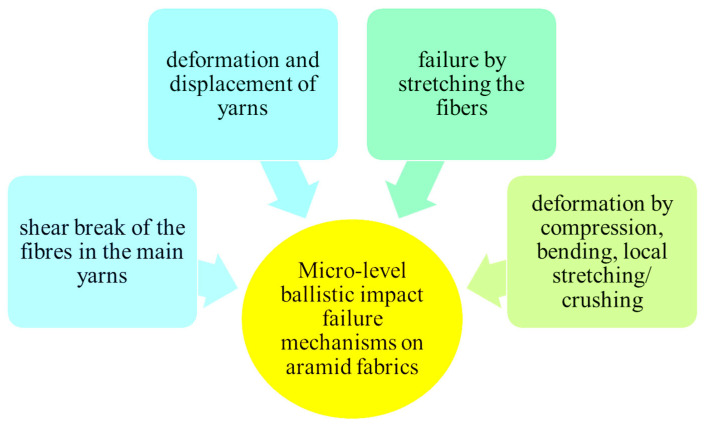
Micro-level failure mechanisms upon ballistic impact on tested aramid fabric.

**Figure 10 polymers-16-01920-f010:**
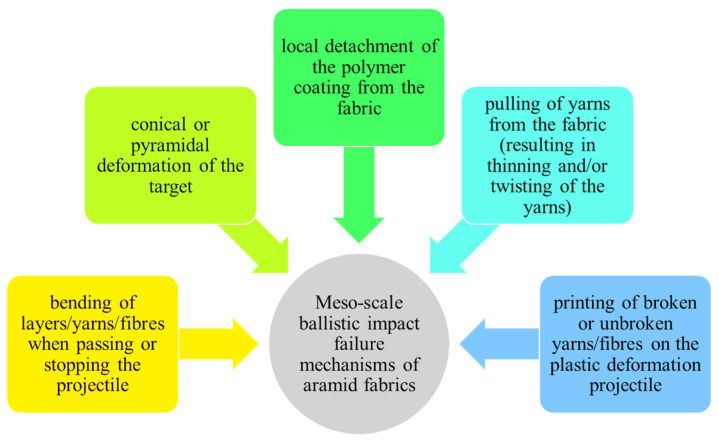
Meso-scale failure mechanisms upon ballistic impact on tested aramid fabric.

**Figure 11 polymers-16-01920-f011:**
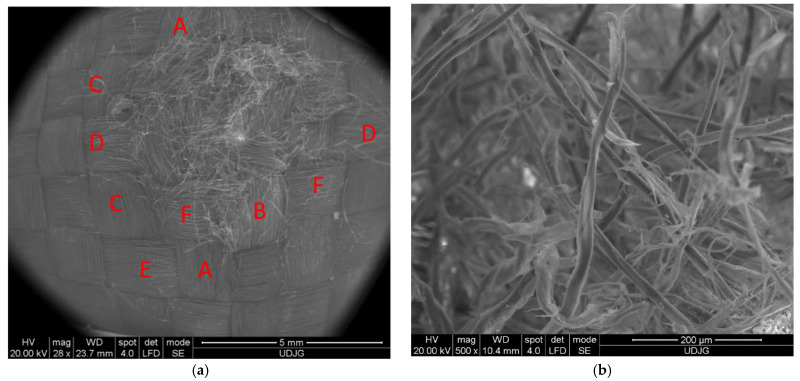
Back of layer 1 from the Twaron CT736 32-layer panel’s modes of fiber failure: (**a**) back view; (**b**) detail.

**Figure 12 polymers-16-01920-f012:**
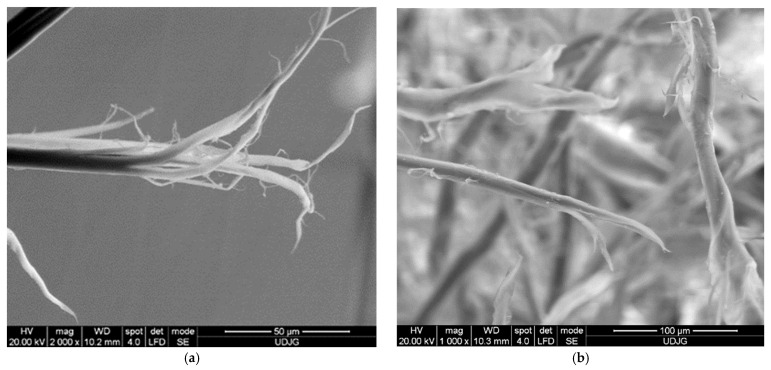
Fibrillation of aramid fibers, after ballistic impact test: (**a**) 32 layers Twaron CT736, fire B, layer 2, front view; (**b**) 32 layers Twaron CT736, fire B, layer 1, front view; (**c**) 32 layers Twaron CT736, fire B, layer 2, front view; (**d**) 32 layers Twaron CT736, fire B, layer 13, front view.

**Figure 13 polymers-16-01920-f013:**
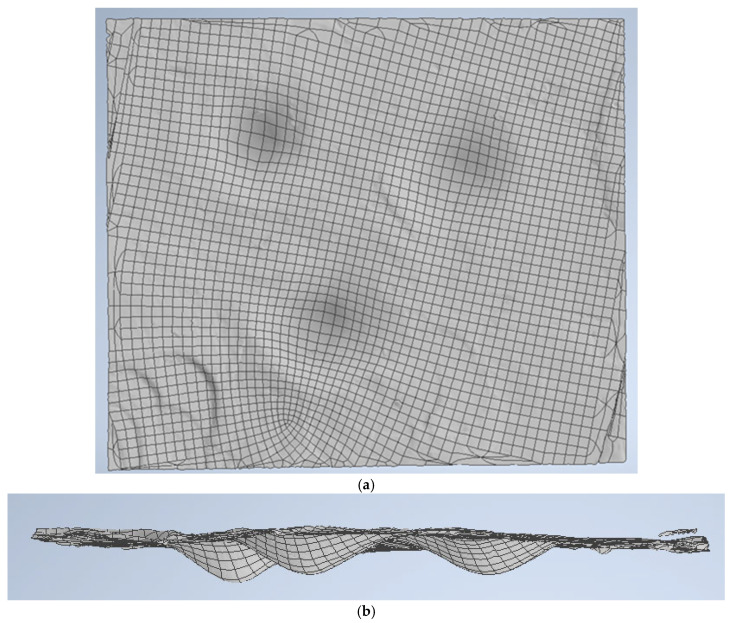
Three-dimensional view of the bearing material after impact of the 32-layer Twaron CT736 panel; (**a**) front view of the bearing material after impact of the panel with three projectiles; (**b**) cross-section.

**Figure 14 polymers-16-01920-f014:**
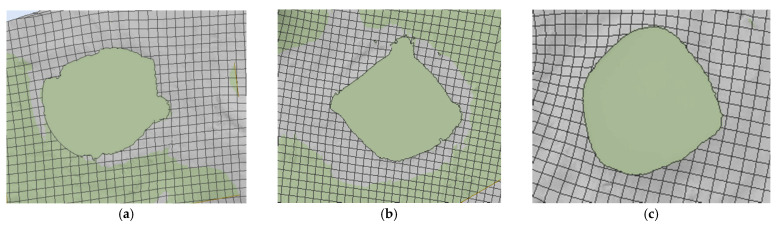
Front view of the contour of the indentation in the reference plane of the backing material (**a)** fire A; (**b**) fire B; (**c**) fire C.

**Figure 15 polymers-16-01920-f015:**
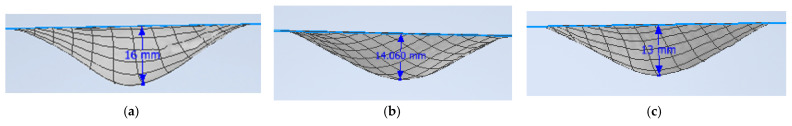
Three-dimensional view with the surface delineation of the indentation in a plane containing the undeformed surface of the support material, vertical section (**a**) fire A; (**b**) fire B; (**c**) fire C.

**Figure 16 polymers-16-01920-f016:**
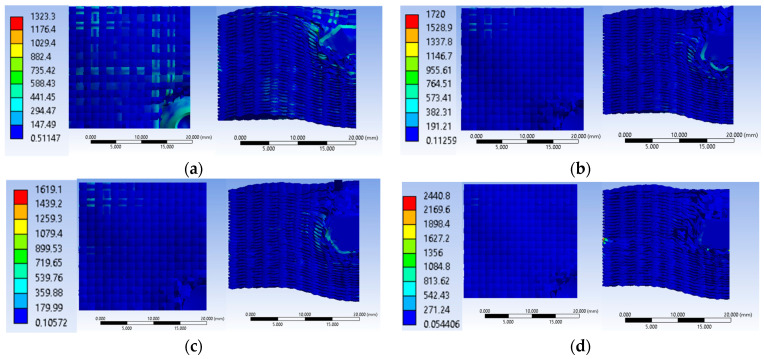
Equivalent stress: (**a**) v_0_ = 414 m/s, yield strength = 3600 MPa; (**b**) v_0_ = 414 m/s, Yield strength = 3000 MPa; (**c**) v_0_ = 422 m/s, yield strength = 3000 MPa; (**d**) v_0_ = 428 m/s, yield strength = 3000 MPa.

**Figure 17 polymers-16-01920-f017:**
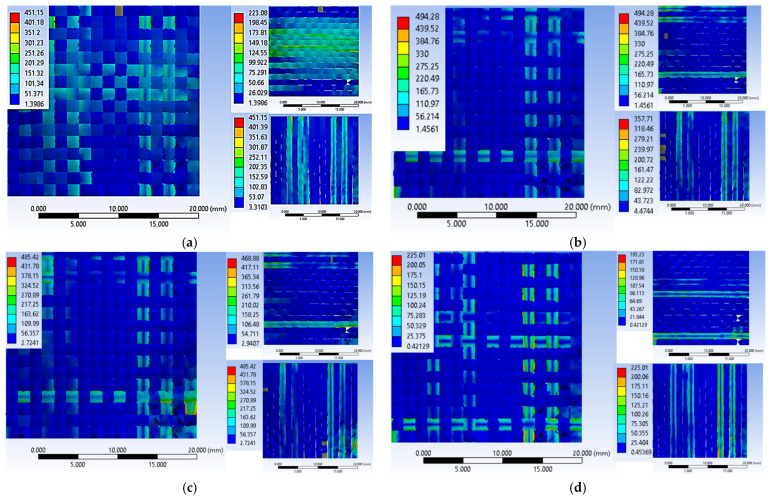
The last layer destroyed, Variant 1: (**a**) v_0_ = 414 m/s, yield strength = 3600 MPa; (**b**) v_0_ = 414 m/s, yield strength = 3000 MPa; (**c**) v_0_ = 422 m/s, yield strength = 3000 MPa; (**d**) v_0_ = 428 m/s, yield strength = 3000 MPa.

**Figure 18 polymers-16-01920-f018:**
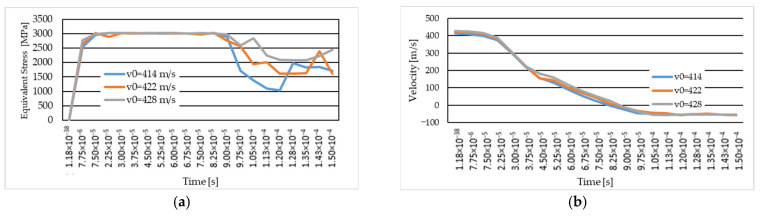
(**a**) Equivalent stress for the three variants studied (in MPa); (**b**) velocity for the three variants studied (in m/s).

**Figure 19 polymers-16-01920-f019:**
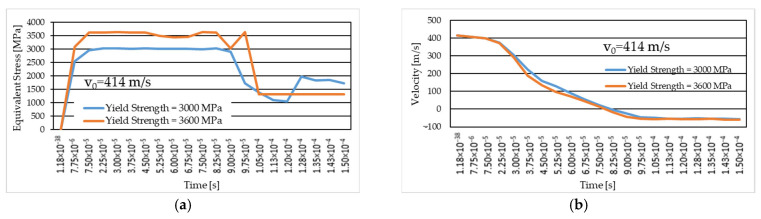
The case with initial projectile velocity of 414 m/s (different thread yield strength): (**a**) equivalent stress (in MPa); (**b**) velocity (in m/s).

**Figure 20 polymers-16-01920-f020:**
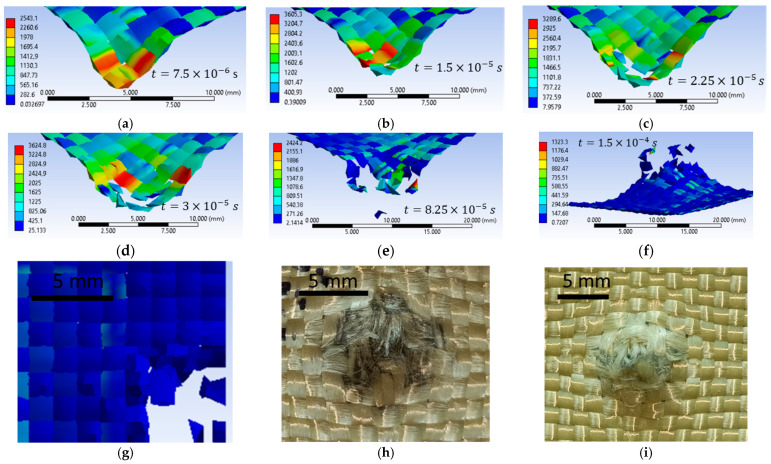
Layer 1, initial projectile velocity of 414 m/s, 3600 MPa yield strength (**a**–**g**) numerical simulation; (**h**,**i**) laboratory test.

**Figure 21 polymers-16-01920-f021:**
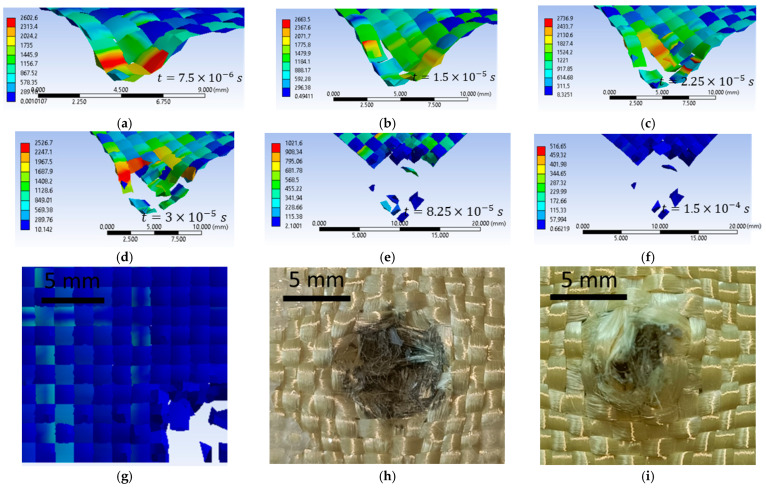
Layer 1, initial projectile velocity of 422 m/s, 3000 MPa yield strength (**a**–**g**) numerical simulation; (**h**,**i**) laboratory test.

**Figure 22 polymers-16-01920-f022:**
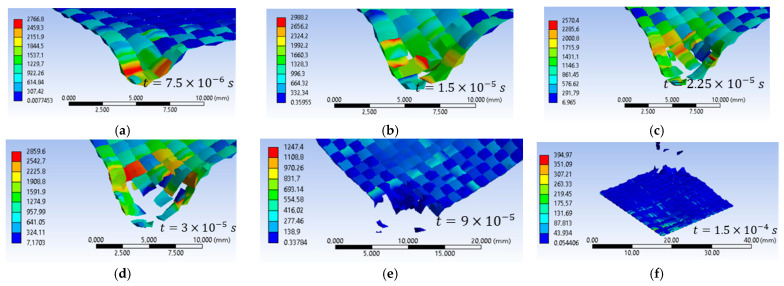
Layer 1, initial projectile velocity of 428 m/s, 3000 MPa yield strength (**a**–**g**) numerical simulation; (**h**,**i**) laboratory test.

**Figure 23 polymers-16-01920-f023:**
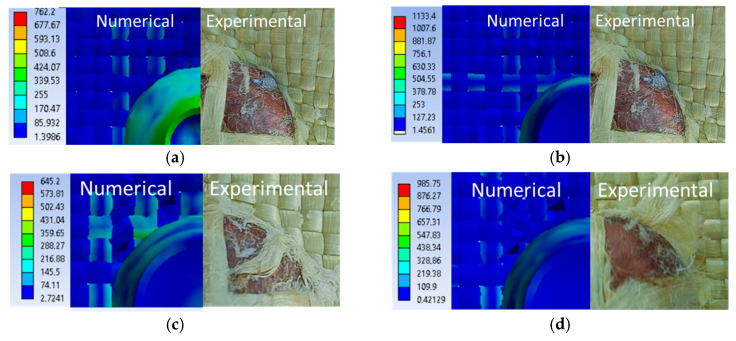
The last damaged layer from the numerical simulation and laboratory test, Variant 1: (**a**) v_0_ = 414 m/s, yield strength = 3600 MPa; (**b**) v_0_ = 414 m/s, yield strength = 3000 MPa; (**c**) v_0_ = 422 m/s, yield strength = 3000 MPa; (**d**) v_0_ = 428 m/s, yield strength = 3000 MPa.

**Figure 24 polymers-16-01920-f024:**
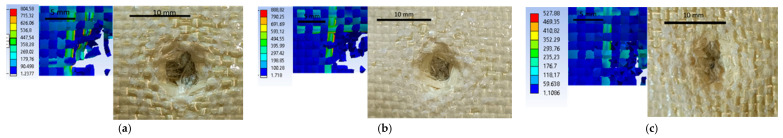
Qualitative aspect, variants 2 (**a**) layer 3; (**b**) layer 5; (**c**) layer 10.

**Figure 25 polymers-16-01920-f025:**
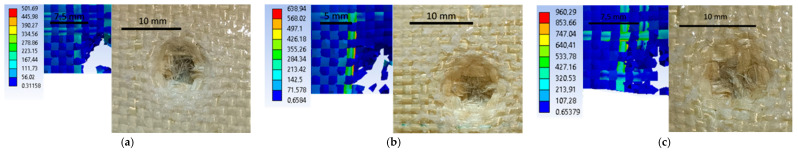
Qualitative aspect, variants 3 (**a**) layer 3; (**b**) layer 5; (**c**) layer 10.

**Figure 26 polymers-16-01920-f026:**
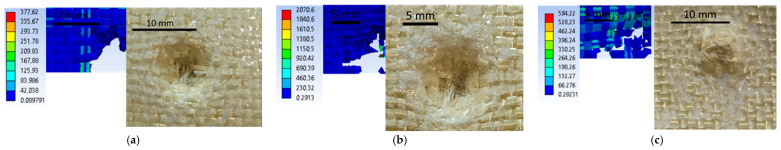
Qualitative aspect, variants 4 (**a**) layer 3; (**b**) layer 5; (**c**) layer 10.

**Table 1 polymers-16-01920-t001:** Twaron CT736 fabric characteristics [51].

Linear Density [dtex_nom_]	Twaron-Type	Set [per 10 cm]		Areal Density [g/m^2^]	Thickness [mm]	Minimum Breaking Strength [N/5 cm × 1000]	
		Warp	Weft			Warp	Weft
1680f1000	2000	127	127	410	0.62	15.5	16.60

**Table 2 polymers-16-01920-t002:** Thickness of tested panel.

Panel	Order and Number of Layers of Material	Weight of Material Layers [g]	Total Weight [g]
32 layers of Twaron CT736 fabric	16 layers Twaron CT736	1202 (±1)	2421
	16 layers Twaron CT736	1219 (±1)	

**Table 3 polymers-16-01920-t003:** The material properties of the yarn.

Property	Value	Unit
Density	1440	Kg/m^3^
Isotropic Elasticity		
Young’s Modulus	91000	MPa
Poisson’s Ratio	0.35	-
Bulk Modulus	1.0111 × 10^11^	Pa
Shear Modulus	3.3704 × 10^10^	Pa
Bilinear Isotropic Hardening		
Yield Strength	3000	MPa
Tangent Modulus	19,000	MPa
Plastic Strain Failure		
Equivalent Plastic Strain EPS	0.031	-

**Table 4 polymers-16-01920-t004:** The material properties of the projectile jacket.

Property	Value	Unit
Density	8300	Kg/m^3^
Isotropic Elasticity		
Young’s Modulus	1.17 × 10^5^	MPa
Poisson’s Ratio	0.34	-
Bulk Modulus	1.2188 × 10^11^	Pa
Shear Modulus	4.3657 × 10^10^	Pa
Bilinear Isotropic Hardening		
Yield Strength	70	MPa
Tangent Modulus	1150	MPa
Plastic Strain Failure		
Maximum Equivalent Plastic Strain EPS	1	-

**Table 5 polymers-16-01920-t005:** The material properties of the projectile core.

Property	Value	Unit
Density	11,340	Kg/m^3^
Isotropic Elasticity		
Young’s Modulus	16,000	MPa
Poisson’s Ratio	0.44	-
Bulk Modulus	4.4444 × 10^11^	Pa
Shear Modulus	5.5556 × 10^9^	Pa
Johnson–Cook Strength		
Initial Yield Strength	24	MPa
Hardening Constant	300	MPa
Hardening Exponent	1	-
Strain Rate Constant	0.1	-
Thermal Softening Exponent	1	-
Melting Temperature	760	K
Reference Strain Failure	1	(/s)
Plastic Strain Failure		
Maximum Equivalent Plastic Strain EPS	1	-

**Table 6 polymers-16-01920-t006:** Values of the depth of identification (BFS) in the support material.

Sample	Depth of Indentation/Back Face Signature—Measurement with a Screwdriver BFS [mm]	Depth of Identification/Back Face Signature—3D Scanning BFS_(3D)_ [mm]	ΔBFS = BFS-BFS_(3D)_ [mm]	ΔBFS% = (BFS-BFS_(3D)_) × 100/BFS [%]
Fire A	17 (±0.1)	16.000 (±0.1)	1.000	5.88
Fire B	16 (±0.1)	14.060 (±0.1)	1.940	12.13
Fire C	14 (±0.1)	13.000 (±0.1)	1.000	7.14

## Data Availability

All data having references could be publicly accessed.

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
