# Peer review of "Experimental and Numerical Analysis of the Damage Mechanism of an Aramid Fabric Panel Engaged in a Medium-Velocity Impact"

_polymers, 2024, doi:10.3390/polym16131920_

Round 1
Reviewer 1 Report
Comments and Suggestions for Authors
The paper is interesting and has two methods to validate the damage mechanism of aramid fabric panels. However, the overall writing style and content are required to be revised, please refer to the below comments.
1. Title: the specimen of your sample is a kind of laminate or panel, it is not clearly defined, aramid fabric is the material type, which is not your specimen shape. In addition, impact has many levels, such as low-velocity, high-velocity, etc. Please specify the impact type.
2. Type of this paper, it is Research Article.
3. Abstract: FMJ? what is this standard for? Methods are not mentioned. Findings are not highlighted, it needs to involve the impact load versus time curve and damage mechanism. please also provide the experimental and numerical results.
4. Keywords: it is suggested to add two or three keywords, such as damage mechanism, impact resistance,
5. Introduction: It is not well reviewed on the impact of laminate with different fabric types. Moreover, the damage mechanism is not clearly reviewed. For ballistic test parameters, it is better to discuss the required parameters. Last paragraph, the research scope and problem statement are not discussed well. Please highlight it. Line 60: BFS is not put in the right place, is it back face signature?
6. Materials and methods section: For Table 2, it is not included all specimens with different layers. For Figure 1 (a), the reviewer focuses on the woven type, it needs to use a microscope rather than SEM, please show the cross-section view of fabric yarn. Finite element modelling is another method, it needs to be added in this section, please remove some sections of analysis of numerical simulation from Line 285. The experimental setup and how the author collects the data are missing in this section.
6. Results and discussion: it is suggested to add the experimental and numerical results in this section. Figure 2, why back view is gray and front view yellow? For the results, it needs to be compared with some references to discuss and support your work.
7. In Figure 4, the bullet needs to move out and discuss the the back view of the specimen.
8. Figure 6, it is required to observe the microscope damage in the image with a label rather than A, B, and C, it is better to label with damage type.
9. Figure 7 and Figure 8 are suggested to discuss with the observation image of this study, the brief failure mechanism is understood, and it can move to the literature review rather result and discussion section,
10. Figure 10, it is labelled in the SEM image with failure mechanism rather than A, B, C,...
11. Table 3, for the FE, it is considered to discuss the impact area and indentation displacement, therefore, Table 3 is not a significant parameter and can be removed.
12. 3.1 The model can be moved to a Method with boundary conditions, impact speed, etc. In results and discussion, it focus on the FE findings.
13. There are many layers of specimens, and it is suggested to cut them in half to observe the cross-section view of specimens, which can validate the FE results. Compared with FE, which data the author is compared?impact speed, displacement, or strength? how about the FE data?
14. In Figure 23, the bullet is not removed, please remove and compare again. Moreover, the dimension of the bullet is not shown in this paper, it should be added.
15. Conclusion: It is too long without main findings and data. Please rewrite the sentences.
Comments on the Quality of English LanguageThe English of this manuscript requires moderate proofreading before revision submission.
Author Response
|
Comments 1: Title: the specimen of your sample is a kind of laminate or panel, it is not clearly defined, aramid fabric is the material type, which is not your specimen shape. In addition, impact has many levels, such as low-velocity, high-velocity, etc. Please specify the impact type.
|
|
Response 1: Thank you for pointing this out. We agree with this comment. Therefore, we added the word panel in the title and also specified the type of impact. |
|
|
|
Comments 2: Type of this paper, it is Research Article. |
|
Response 2: Thank you for pointing this out. We agree with this comment.
|
|
Comments 3: Abstract: FMJ? what is this standard for? Methods are not mentioned. Findings are not highlighted, it needs to involve the impact load versus time curve and damage mechanism. please also provide the experimental and numerical results.
Response 3: Thank you for pointing this out. We agree with this comment. We have explained in brackets what FMJ means and what this standard is for. (line 8, line10 and line 11). Numerical and experimental results have been added in the abstract (line 17-21, line 22-25)
|
|
Comments 4: Keywords: it is suggested to add two or three keywords, such as damage mechanism, impact resistance, Response 4: Thank you for pointing this out. We agree with this comment. We have modified the keywords (line 29)
|
|
Comments 5: Introduction: It is not well reviewed on the impact of laminate with different fabric types. Moreover, the damage mechanism is not clearly reviewed. For ballistic test parameters, it is better to discuss the required parameters. Last paragraph, the research scope and problem statement are not discussed well. Please highlight it. Line 60: BFS is not put in the right place, is it back face signature? Response 5: Thank you for pointing this out. We agree with this comment. there is only one type of fabric (Twaron CT736) analyzed, a panel consisting of two panels with 16 layers of fabric, fastened in the corners, and the two panels were then fastened with textile purposes. We added the characteristics of the aramid fiber (line 54-67) as well as the parameters required for a protection system (line 73-83). The destruction mechanisms identified in this work are given in Figure 9 and Figure 10 at micro scale and meso scale. I clarified the purpose of the research and the statement of the problem (page number 3, line number 113-125
|
|
Comments 6: Materials and methods section: For Table 2, it is not included all specimens with different layers. For Figure 1 (a), the reviewer focuses on the woven type, it needs to use a microscope rather than SEM, please show the cross-section view of fabric yarn. Finite element modelling is another method, it needs to be added in this section, please remove some sections of analysis of numerical simulation from Line 285. The experimental setup and how the author collects the data are missing in this section. |
|
Response 6: Thank you for pointing this out. We agree with this comment. the panel is made of a single material. unfortunately I don't have a picture of the cross section of the thread. I have moved the finite element modelling to the Materials and methods section (page number: 4-7, line 165-216). I specify the method of data collection (page number 3, line 146-148. |
Comments 6: Results and discussion: it is suggested to add the experimental and numerical results in this section. Figure 2, why back view is gray and front view yellow? For the results, it needs to be compared with some references to discuss and support your work.
Response 6: Thank you for pointing this out. We agree with this comment. Figure 2 shows photographs of the panel and images obtained using 3D scanning. We have changed the title of the figure for better understanding (page number: 8 and line 260-261). We have added references in the text to support my findings (page number: 7- 8 and line 239-257; page number 10-11, line 322-343). Line 161-163
Comments 7. In Figure 4, the bullet needs to move out and discuss the the back view of the specimen.
Response : Thank you for pointing this out. The projectile cannot be moved because it is stuck to the fabric.
Comments 8: Figure 6, it is required to observe the microscope damage in the image with a label rather than A, B, and C, it is better to label with damage type.
Response 8: Thank you for pointing this out. We have chosen this variant of lettering on the figure in order not to encumber the figure and for clearer visibility.
Comments 9: Figure 7 and Figure 8 are suggested to discuss with the observation image of this study, the brief failure mechanism is understood, and it can move to the literature review rather result and discussion section,
Response 9: Thank you for pointing this out. We made these diagrams to group the fabric breaking mechanisms at the yarn level and at the fiber level. We considered it important to mention schematically the breaking mechanisms of this fabric. Based on the figures, these mechanisms have been discussed in the text.
Comments 10: Figure 10, it is labelled in the SEM image with failure mechanism rather than A, B, C,...
Response 10: Thank you for pointing this out. We have chosen this variant of lettering on the figure in order not to encumber the figure and for clearer visibility.
Comments 11: Table 3, for the FE, it is considered to discuss the impact area and indentation displacement, therefore, Table 3 is not a significant parameter and can be removed.
Response 11: Thank you for pointing this out. We agree with this comment. Therefore, we deleted the table.
Comments 12: 3.1 The model can be moved to a Method with boundary conditions, impact speed, etc. In results and discussion, it focus on the FE findings.
Response 12: Thank you for pointing this out. We agree with this comment. we have moved ''The model'' into ‘’Materials and Methods’’ (page number: 4-7, line 165-216).
Comments 13: There are many layers of specimens, and it is suggested to cut them in half to observe the cross-section view of specimens, which can validate the FE results. Compared with FE, which data the author is compared?impact speed, displacement, or strength? how about the FE data?
Response 13: Thank you for pointing this out. We agree with this comment. Because the samples were cut by hand with scissors and because the thread is made of fibers, when I cut it in half, the fabric falls apart and around impact I would have a concrete picture of what happened to the fibers in the thread. I analyzed layer by layer, and with the help of a needle I analyzed if the thread was broken completely some of the fibers in the thread were broken. The results of the numerical simulation were compared in terms of qualitative appearance and number of layers destroyed by the projectile.
Comments 14: In Figure 23, the bullet is not removed, please remove and compare again. Moreover, the dimension of the bullet is not shown in this paper, it should be added.
Response 14: Thank you for pointing this out. We agree with this comment. But unfortunately, the projectile got caught on the fabric, and if I had pulled it to detach it from the fabric, I would have broken the fibers attached to it.
Comments 15: Conclusion: It is too long without main findings and data. Please rewrite the sentences.
Response 15: Thank you for pointing this out. We agree with this comment. Therefore, we have rewritten the conclusions (page number: 22-23, paragraph 1-6, and line 592-621).
Reviewer 2 Report
Comments and Suggestions for Authors
Cristian Muntenita et al., in their manuscript entitled "Experimental and numerical analysis of the damage mechanism of aramid fabric at impact," analyzed the ballistic impact behavior of a panel made of Twaron CT736 fabric with a 9 mm FMJ projectile. By examining the macrophotographs of the material impact, they studied the material performance and failure mechanisms. However, some of the limitations listed below were observed:
- The manuscript contains many typos and spelling mistakes. They should be corrected.
- The novelty of this work is absent. The authors should explain why they chose Twaron CT736 and why it is better than other well-known high-performance synthetic fibers. Moreover, in the introduction, the authors should add a short literature review about the properties of the real protection systems, preparation materials and methods, cons, and pros. This review would be important to better compare the obtained results with other findings.
- The authors should explain why they prepared a 32-layer panel.
- The authors should add a short explanation about the NIJ 0101.06 standard.
- The resolution of all presented figures and their legends should be improved.
- The authors should add the explanation of images (c) and (d) to the title of Figure 2.
- The authors wrote, “Twaron CT736 fabric, with a more flexible structure, can better absorb and dissipate the kinetic energy of the projectile, limiting fiber breakage and peeling of polymer coatings.” However, they didn’t provide scientific evidence or an investigation.
- The method of preparation of all samples and the applied conditions for the scanning electron microscope (SEM) test should be added.
- The abbreviation “PVB” should have been explained in the manuscript text when it first appeared.
- The authors should add the related references to figures 7 and 8.
- The authors should add the size of the volume of the indentation in Table 3 (mm3, cm3, etc.).
- The standard deviations for all presented values and numbers should be added.
- The model presented in Section 3 (which should be 4 and not 3) needs to be compared with other similar ones in order to better understand its accuracy.
- The scientific discussion in the section “Validation of the numerical simulations” is very weak; the authors should discuss their model validation in detail. Moreover, which value was obtained for the difference between the numerical simulations and laboratory tests?
- In conclusion, the authors wrote, “Poor adhesion between the substrates and the composite fabric." How did the authors determine the adhesion strength?
- In conclusion, the authors wrote, “From the numerical analysis, we can conclude that if the thread yields strength increases, the number of layers with broken threads decreases. Following impact, the yarn yield strength decreases. At the same time, a small variation in the initial velocity of the projectile does not influence the number of destroyed layers." Unfortunately, the presented results couldn’t confirm this finding.
- In general, the article is technical and descriptive; the main lack is the weak scientific discussion.
- The cited references should be written according to MDPI style, with their DOIs added.
Moderate editing of the English language is required.
Author Response
|
Comments 1: The manuscript contains many typos and spelling mistakes. They should be corrected. |
|
Response 1: Thank you for pointing this out. We agree with this comment. I have revised the text. I have added to the text what you suggested.
|
|
Comments 2: The novelty of this work is absent. The authors should explain why they chose Twaron CT736 and why it is better than other well-known high-performance synthetic fibers. Moreover, in the introduction, the authors should add a short literature review about the properties of the real protection systems, preparation materials and methods, cons, and pros. This review would be important to better compare the obtained results with other findings. |
|
Response 2: Thank you for pointing this out. We agree with this comment. I have added to the text what you suggested. (Page number 2, paragraph 2 and 3, line 54-67; page number 2, paragraph 5, line 73-84; page number 3, paragraph 3 and 4, line 113-125; page number 5, paragraph 2 and 3, line 185-189; page number 7, paragraph 2, line 224-225; page number 7-8 paragraph 5,6,7,8, line 240-258; page number 10, paragraph 4, line 323-326; page number 11, paragraph 2 ,3 and 4. Line 329-344).
Comments 3: The authors should explain why they prepared a 32-layer panel. Response 3: Thank you for pointing this out. We agree with this comment. I have added to the text what you suggested (page number 3, paragraph 3 and 4, line 113-125).
Comments 4: The authors should add a short explanation about the NIJ 0101.06 standard. Response 4: Thank you for pointing this out. We agree with this comment. I have added to the text what you suggested (page number 2, paragraph 7, line 89-96)
Comments 5: The resolution of all presented figures and their legends should be improved. Response 5: Thank you for pointing this out. We agree with this comment. We tried to redo the figures, but unfortunately, we didn't get better results, a solution would be to enlarge them, but I don't know if it's good.
Comments 6: The authors should add the explanation of images (c) and (d) to the title of Figure 2. Response 6: Thank you for pointing this out. We agree with this comment. I added the title of the figures (page number 8, line 261-262)
Comments 7: The authors wrote, “Twaron CT736 fabric, with a more flexible structure, can better absorb and dissipate the kinetic energy of the projectile, limiting fiber breakage and peeling of polymer coatings.” However, they didn’t provide scientific evidence or an investigation. Response 7: Thank you for pointing this out. We agree with this comment. I have rephrased this sentence and added work to support my idea (page number 11, paragraph 1, 2, 3 and 4, line 327-344).
Comments 8: The method of preparation of all samples and the applied conditions for the scanning electron microscope (SEM) test should be added. Response 8: Thank you for pointing this out. We agree with this comment. We have added in the text what you suggested (page number 4, paragraph 2, line 162-164)
Comments 9: The abbreviation “PVB” should have been explained in the manuscript text when it first appeared. Response 9: Thank you for pointing this out. We agree with this comment. We have added in the text what you suggested (page number 3, paragraph 4, line 121; page number 11, paragraph 5, line 352; page number 22, paragraph 1, line 593-594). Because we didn't get some exact dates about when the PVB first appeared, we thought we shouldn't give data about something we're not sure about.
Comments 10: The authors should add the related references to figures 7 and 8. Response 10: Figures 7 and 8 were created by us with the breaking mechanisms that were identified in this study. We have added text to the titles of the figures to make it clear that they were created by us ( page number 12, line 360 and 362).
Comments 11: The authors should add the size of the volume of the indentation in Table 3 (mm3, cm3, etc.). Response 11: Thank you for pointing this out. We agree with this comment. I have added the unit of measurement in the text
Comments 12: The standard deviations for all presented values and numbers should be added. Response 12: Thank you for pointing this out. We agree with this comment. I have added in the text the deviations (page number 4, line 151)
Comments 13: The model presented in Section 3 (which should be 4 and not 3) needs to be compared with other similar ones in order to better understand its accuracy. Response 13: Thank you for pointing this out. We agree with this comment. I have added what you suggested (page number 5, paragraph 2 and 3, line 185-189).
Comments 14: The scientific discussion in the section “Validation of the numerical simulations” is very weak; the authors should discuss their model validation in detail. Moreover, which value was obtained for the difference between the numerical simulations and laboratory tests? Response 14: Thank you for pointing this out. We validated the appearance of the model by the validated appearance (images) and the number of broken layers.
Comments 15: In conclusion, the authors wrote, “Poor adhesion between the substrates and the composite fabric." How did the authors determine the adhesion strength? Response 15: Thank you for pointing this out. We assumed that the PVB substrate had peeled off due to poor adhesion with the fabric.
Comments 16: In conclusion, the authors wrote, “From the numerical analysis, we can conclude that if the thread yields strength increases, the number of layers with broken threads decreases. Following impact, the yarn yield strength decreases. At the same time, a small variation in the initial velocity of the projectile does not influence the number of destroyed layers." Unfortunately, the presented results couldn’t confirm this finding. Response 16: Thank you for pointing this out. We agree with this comment. We have rephrased the conclusions.
Comments 17: In general, the article is technical and descriptive; the main lack is the weak scientific discussion. Response 17: Thank you for pointing this out. We agree with this comment. we have tried with your suggestions to improve the work as much as possible.
Comments 18: The cited references should be written according to MDPI style, with their DOIs added. Response 18: Thank you for pointing this out. We agree with this comment. We added what you suggested in the text.
|
Round 2
Reviewer 2 Report
Comments and Suggestions for Authors
The authors made acceptable improvements to the manuscript. However, some of the limitations listed below were observed:
- The manuscript still contains many typos and spelling mistakes. They should be corrected.
- The novelty of this work is also still absent. Based on the added information (P.3, lines 113–125), the 32-layer Twaron CT736 fabric panel has already been investigated.
- The authors didn't provide a scientific explanation for comment 7 (Comments 7: The authors wrote, “Twaron CT736 fabric, with a more flexible structure, can better absorb and dissipate the kinetic energy of the projectile, limiting fiber breakage and peeling of polymer coatings.” However, they didn’t provide scientific evidence or an investigation.)
Moderate editing of the English language is required.
Author Response
|
Comments 1: The manuscript still contains many typos and spelling mistakes. They should be corrected. |
|
Response 1: Thank you for pointing this out. We agree with this comment. The paper has been revised as indicated.
|
|
Comments 2: The novelty of this work is also still absent. Based on the added information (P.3, lines 113–125), the 32-layer Twaron CT736 fabric panel has already been investigated. Response: Thank you for pointing this out. We agree with this comment. We have made the corresponding changes. What we meant in the paper is that the fabric was staple tested, the same number of layers and we impact tested. (page number 3, line 114-121)
Comments 3: The authors didn't provide a scientific explanation for comment 7 (Comments 7: The authors wrote, “Twaron CT736 fabric, with a more flexible structure, can better absorb and dissipate the kinetic energy of the projectile, limiting fiber breakage and peeling of polymer coatings.” However, they didn’t provide scientific evidence or an investigation.) Response: Thank you for pointing this out. We agree with this comment. I deleted this phrase.
|